# Spontaneous spinal cord infarctions: a systematic review and pooled analysis protocol

Victor Gabriel El-Hajj [1,2], Vasilios Stenimahitis,[2,3] Maria Gharios,[2]
Omar Ali Mahdi,[2] Adrian Elmi-Terander [4,5] Erik Edström[2,4]

¹Department of Neurosurgery, Karolinska University Hospital, Stockholm, Sweden
²Department of Clinical Neuroscience, Karolinska Institutet, Stockholm, Sweden
³Department of Neurology, Karolinska University Hospital, Stockholm, Sweden
⁴Stockholm Spine Center, Löwenströmska Hospital, Upplands-Väsby, Sweden
⁵Department of Surgical Sciences, Uppsala University, Uppsala, Sweden

**Correspondence to**
Mr Victor Gabriel El-Hajj;
victor.gabriel.elhajj@stud.ki.se

## ABSTRACT

**Introduction** Spinal cord infarction (SCInf) is a rare ischaemic event that manifests with acute neurological deficits. It is typically classified as either spontaneous, defined as SCInf without any inciting event, or periprocedural, which typically occur in conjunction with vascular surgery with aortic manipulations. While periprocedural SCInf has recently been the subject of intensified research, especially focusing on the primary prevention of this complication, spontaneous SCInf remains less studied.

**Methods and analysis** Electronic databases, including PubMed, Web of Science and Embase, will be searched using the keywords "spinal cord", "infarction", "ischemia" and "spontaneous". The search will be set to provide only English studies published from database inception. Editorials, letters and reviews will also be excluded. Reference lists of relevant records will also be searched. Identified studies will be screened for inclusion, by one reviewer in the first step and then three in the next step to decrease the risk of bias. The synthesis will address several topics of interest including epidemiology, presentation, diagnostics, treatment strategies, outcomes and predictors. The review aims to gather the body of evidence to summarise the current knowledge on SCInf. This will lead to a better understanding of the condition, its risk factors, diagnosis and management. Moreover, the review will also provide an understanding of the prognosis of patients with SCInf with respect to neurological function, quality of life and mortality. Finally, this overview of the literature will allow the identification of knowledge gaps to help guide future research efforts.

**Ethics and dissemination** Ethics approval was not required for our review as it is based on existing publications. The final manuscript will be submitted to a peer-reviewed journal.

## STRENGTHS AND LIMITATIONS OF THIS STUDY

⇒ Our wide search strategy and limited set of exclusion criteria allow many studies to be included, ensuring adequate coverage of the topic and correct identification of knowledge gaps.
⇒ By providing a comprehensive synthesis of the body of evidence, the data can form the basis for management guidelines and future research efforts.
⇒ We suspect that the quality of data does not suffice to perform a meta-analysis, consequently limiting the level of evidence that can be attained.

different degrees of severity,[5 8] bladder, bowel or autonomic system dysfunction.[3 9 10] Moreover, rapid neurological deterioration is not uncommon and often suggestive of a worse prognosis.[5 11]

SCInf may occur spontaneously or in a periprocedural or traumatic setting.[11 12] Spontaneous SCInf is defined as an ischaemic event occurring without any identifiable inciting event, while periprocedural SCInf are the result of an iatrogenic or traumatic disruption of blood flow through spinal arteries originating in the aorta.[13–16] Regardless of the aetiology, there are no definitive diagnostic criteria for SCInf. To address this gap, Zalewski *et al* proposed a diagnostic algorithm that relies on clinical presentation, imaging and other complementary diagnostic methods.[11] SCInf was divided into either spontaneous or periprocedural, and subclassified, based on the specificity of the diagnostic findings into definite, probable or possible (figure 1).

Spontaneous SCInf has been hypothesised to result from vascular disease processes analogous to those in cerebral stroke. However, the pathophysiology of spontaneous SCInf has been the subject of significantly less research.[17] Nonetheless, the presence of vascular risk factors such as hypertension, hyperlipidaemia, diabetes and smoking is well documented in patients with spontaneous

## INTRODUCTION

Spinal cord infarction (SCInf) is a rare ischaemic occurrence representing only a fraction of all ischaemic strokes (1%)[1–4] and approximately 6% of all acute myelopathies.[5] The presentation of SCInf is characterised by the rapid onset of symptoms that reflect the affected spinal cord segment.[6] These symptoms include, but are not limited to, back pain,[7] sensory or motor deficits with

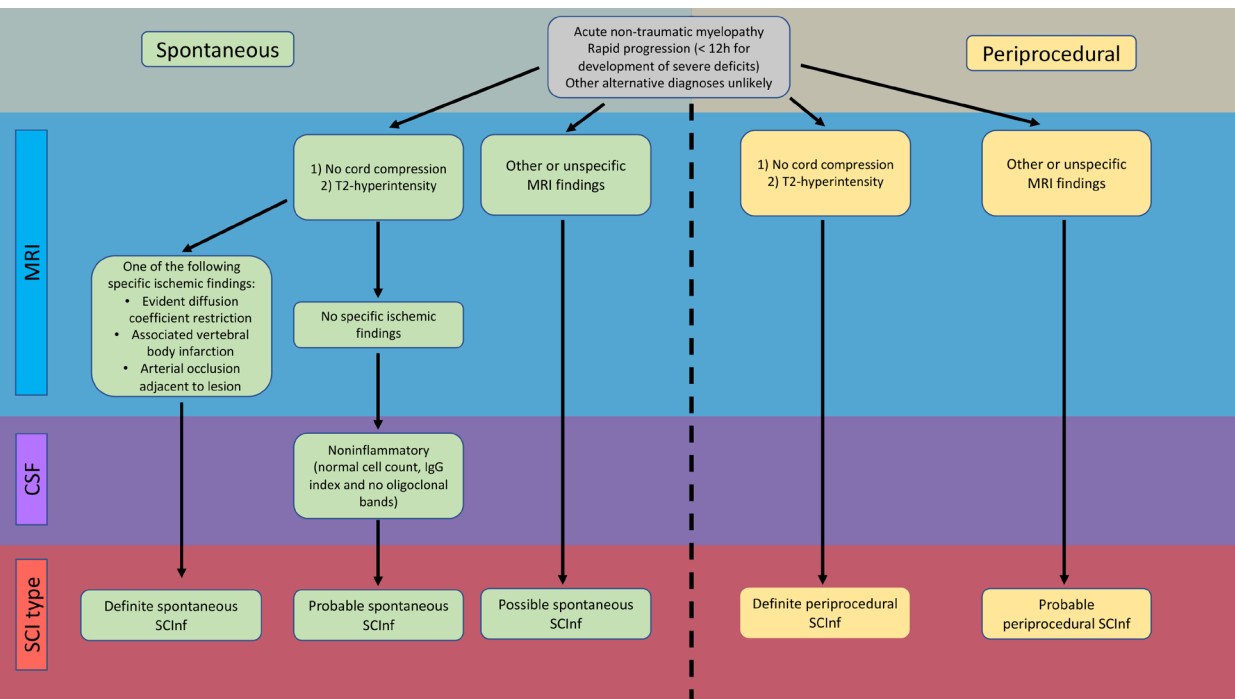

**Figure 1** Flow chart illustrating the categorisation of SCInf based on the certainty of diagnostic findings, as defined by Zalewski *et al.*[11] CSF, cerebrospinal fluid; SCInf, spinal cord infarction.

SCInf.[11] However, more research is needed to clarify causal relationships.

While it is easier to diagnose periprocedural SCInf, as they are well-established complications mainly occurring in conjunction with aortic surgery, the diagnosis of spontaneous SCInf, remains more challenging. This complexity derives from several factors, of which the rarity of the condition,[1–4] its overlap with other acute myelopathies,[5 9 11] and the variability in presenting signs and symptoms, are the largest contributors. Additionally, previous studies seem to indicate that up to half of imaging workups in patients with a clinical suspicion of SCInf may return normal[18] and that a spinal cord lesion may not be accurately discernible with T2-weighted imaging during the first 15 hours.[14 16 19] Hence, spontaneous SCInfs require a certain degree of clinical suspicion and expertise in order for a timely diagnosis to be made.[16]

Currently, there are no clear guidelines for the management of SCInf.[15] Measures have been suggested, to prevent the occurrence of periprocedural ones. Although still a matter of debate,[15 20–22] cerebrospinal fluid (CSF) drainage was proposed as a means to reduce intraspinal pressure and thereby enhance perfusion of the spinal cord.[21] In fact, recent reports addressing this topic have emphasised the value of maintaining an adequate spinal cord perfusion pressure, both during and after aortic procedures.[23] In contrast, the management of spontaneous SCInf, mainly relies on the treatment guidelines of ischaemic cerebral stroke and myocardial infarction. Consequently, the mainstay consists of the reduction of cardiovascular risk factors[24] and antiplatelet therapy in eligible patients.[25] In the acute phase, thrombolytic

therapy has been described.[24 26 27] However, the available literature includes only a few cases and consequently evidence to support this treatment strategy is currently lacking. Similarly, the use of corticosteroids early in the course of SCInf has been suggested, but supporting evidence is scarce.[28]

The ability to walk is an essential parameter in the aftermath of spinal cord injury. In SCInf, previous surveys are conflicting regarding the proportion of patients with preserved ambulatory function (walking with or without aids) ranging from 38%[9] to 70%.[18] Considering neurological function at long-term follow-up, previous studies seem to agree that a gradual improvement occurs during an extended period of time after SCInf.[29 30] Analysis of the predictors of unfavourable outcomes revealed multisegment lesions,[4] lower admission ASIA scores[18] and older age[2] as predictors of outcome. Sex has not been identified as an outcome predictor.[18] Although not consistent

**Table 1** SPIDER criteria

| **Sample** | **Any patient** |
| --- | --- |
| Phenomenon of interest | Spontaneous spinal cord infarction |
| Design | Descriptive studies with numeric data |
| Evaluation | Epidemiology, risk factors, treatment and outcomes |
| Research type | Experimental and observational studies |

SPIDER, Sample, Phenomenon of Interest, Design, Evaluation, Research.

**Table 2** Level of evidence based on primary research question by Wright et al[41]

| | Therapeutic studies—investigating the results of treatment | Prognostic studies—investigating the outcome of disease | Diagnostic studies—investigating a diagnostic test |
|---|---|---|---|
| Level I | 1. Good-quality randomised controlled trial<br>2. Systematic review of level-I studies | 1. Prospective study<br>2. Systematic review of level-I studies | 1. Testing of previously developed diagnostic criteria in series of consecutive patients (with universally applied reference 'gold' standard),<br>2. Systematic review of level-I studies |
| Level II | 1. Prospective cohort study<br>2. Poor-quality randomised controlled trial<br>3. Systematic review<br> 1. Level-II studies<br> 2. Nonhomogeneous level-I studies | 1. Retrospective study<br>2. Study of untreated controls from a previous randomised controlled trial,<br>3. Systematic review of level-II studies | 1. Development of diagnostic criteria on basis of consecutive patients (with universally applied reference 'gold' standard),<br>2. Systematic review of level-II studies |
| Level III | 1. Caseßcontrol study<br>2. Retrospective cohort study<br>3. Systematic review of level-III studies | | 1. Study of nonconsecutive patients (no consistently applied reference 'gold' standard)<br>2. Systematic review of level-III studies |
| Level IV | Case series (with no, or historical, control group) | Case series | 1. Case–control study<br>2. Poor reference standard |
| Level V | Expert opinion | Expert opinion | Expert opinion |

through the literature,[30] there seems to be preliminary evidence pointing towards worse outcomes in patients with periprocedural SCInf, in contrast to those with spontaneous SCInf.[2 31]

The lack of definitive guidelines for the diagnosis and management of SCInf, as well as the conflicting evidence present throughout the literature warrants a review to identify and summarise the current knowledge regarding risk factors, diagnosis, management, and outcomes of spontaneous SCInf. The planned systematic review aims to synthesise the relevant knowledge pertaining to this topic as well as to highlight knowledge gaps in need of intensified research efforts. Contrary to periprocedural SCInf where comprehensive systematic reviews have been extensively carried out,[23 32–35] there are to the best of our knowledge none addressing spontaneous SCInf. Instead of the more classic Population, Intervention, Comparison, Outcome criteria, we decide to use the Sample, Phenomenon of Interest, Design, Evaluation, Research typecriteria,[36] which we believe are better suited to the purpose of this review (table 1).

## METHODS AND ANALYSIS
### Study registration
This protocol for an intended systematic review is reported according to the Preferred Reporting Items for Systematic Reviews and Meta-Analyses Protocol (PRISMA-P) statement of 2015.[37] The PRISMA-P checklist is provided asonline supplemental file 1. The systematic review protocol was registered on PROSPERO (registration ID: CED42023393241; registration date: 24 February 2023).

### Patient and public involvement
Patients were not involved in the design or conception of the study.

### Eligibility criteria
#### Inclusion criteria
##### Type of studies
All peer-reviewed and original studies, written in English and available in the PubMed, Embase or Web of Science databases from inception and onwards, will be eligible for inclusion.

##### Type of participant
All patient with spontaneous SCInf will be included, regardless of age, ethnicity and sex.

##### Type of outcome measurements
Epidemiological data such as age, sex and socioeconomic factors, risk factors, diagnosis and management strategies, outcomes and predictors will all be addressed. Furthermore, outcome parameters, including pathological mechanisms, quality of life and mortality, will be explored on sufficient data.

#### Exclusion criteria
Non-original publications such as reviews, editorials and letters to the editor will be disregarded together with conference abstracts and case reports. Studies found in languages other than English will be excluded for practical reasons. Non-spontaneous cases of SCInf occurring after clear inciting events, such as surgery, trauma or hypovolaemic shock will be disregarded and excluded from the analysis. Studies only addressing SCInfs secondary to

vertebral artery dissections will also be excluded, as this topic has specifically been addressed in a previous systematic review.[38]

## Databases and search strategy

An electronic database search will be performed in PubMed, Embase and Web of Science. A filter will be used to exclude non-English studies. To illustrate the process, the preliminary search strategy for each of the databases is provided (online supplemental file 2).

## Study selection

The records retrieved from the different databases will be exported into the Rayyan Software.[39] After deduplication, the records will be screened based on title and abstract by one reviewer, to eliminate records that are plainly irrelevant. This is necessary as an unmanageable number of records is foreseen due to the broad search strategy that will be used. In the next step, three independent and blinded reviewers will be assigned the task of examining the remaining records applying the eligibility criteria based on full-text reading. Potential disagreements will be resolved by discussion with a fourth reviewer. Finally, reference lists of the selected articles will be thoroughly reviewed for any potentially eligible studies that were previously missed. The whole process will be illustrated in a PRISMA flow chart, which will be provided in the final manuscript.

## Data extraction

Data from selected records will be extracted using a predefined extraction template, preliminarily including (1) general information—title, first author, journal, publication year, etc; (2) study characteristics—study type, sample size, follow-up time, etc; (3) patient characteristics and epidemiology—age, sex, spinal segment involved, presenting symptoms and neurological function, etc; (4) diagnosis and treatment characteristics—diagnostic modalities, treatment strategy, etc and (5) outcomes—neurological outcomes, predictors of outcome, quality of life, etc. The collaboration of multiple reviewers will be sought to achieve thorough extraction of the data. The final work will be assessed and cross-checked to prevent any error.

## Assessment of risk of bias

The Oxford Center for Evidence-Based Medicine system,[40] modified by Wright *et al*, will be used to assess evidence levels[41 42] (table 2). The selected articles will be allocated to one of four levels based on methodological quality, since the fifth level (V) pertains to expert opinions which are systematically excluded from our study. Risk of bias will be assessed using the appropriate tools specific to the type of study, as defined by Ma *et al*.[43]

## Data synthesis

After extraction, the data obtained from eligible studies will be systematically presented. Topics of interest to this review are chosen as follows:

1. Patient characteristics: epidemiology and risk factors.
2. SCInf characteristics: spinal cord level affected and associated presenting symptoms.
3. Diagnosis: modalities used.
4. Management: treatment options and strategies adopted.
5. Patient outcomes: neurological outcomes, and predictors, quality of life and mortality.

The synthesis will address each of the mentioned topics in separate sections. In the absence of a satisfactory amount or quality of data, the synthesis will primarily take a narrative form and pool the available evidence to achieve higher power and more reliable information. Meta-analysis will only be performed for research questions where rigorous, homogeneous and sufficient data are available. A similar methodology has been described in several of our previous works.[44–47]

## ETHICS AND DISSEMINATION

Ethics approval is not required for this systematic review as it is based on existing publications. The review is planned to take place between the months of February and August of the year of 2023. We then plan to submit our work to a peer-reviewed journal where the results will be openly available.

**Contributors** VGE-H: conception and design of the work, drafting of the article, critical revision, and final approval of the version to be published. VS: conception and design of the work, drafting of the article, and final approval of the version to be published. MG: conception and design of the work, drafting of the article and final approval of the version to be published. OAM: conception and design of the work, drafting of the article, and final approval of the version to be published. AE-T: guarantor of the review, conception and design of the work, drafting of the article, critical revision, and final approval of the version to be published. EE: conception and design of the work, drafting of the article, critical revision and final approval of the version to be published.

**Funding** AE-T was supported by Region Stockholm (clinical research appointment). None of the other authors received funding.

**Competing interests** None declared.

**Patient and public involvement** Patients and/or the public were not involved in the design, or conduct, or reporting, or dissemination plans of this research.

**Patient consent for publication** Not applicable.

**Provenance and peer review** Not commissioned; externally peer reviewed.

**ORCID iDs**
Victor Gabriel El-Hajj http://orcid.org/0000-0001-9479-761X
Adrian Elmi-Terander http://orcid.org/0000-0002-3776-6136

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
