## [Reviewer comments · BMJ Open]

ARTICLE DETAILS

TITLE (PROVISIONAL)	Spontaneous spinal cord infarctions: a systematic review and pooled analysis protocol
AUTHORS	El-Hajj, Victor Gabriel; Stenimahitis, Vasilios; Gharios, Maria; Mahdi, Omar Ali; Elmi-Terander, Adrian; Edström, Erik

VERSION 1 – REVIEW

REVIEWER	Sellner, Johann Paracelsus Medical University Salzburg
REVIEW RETURNED	28-Jan-2023

GENERAL COMMENTS	The protocol for the systematic literature review on seems to consider all relevant aspects. The analysis fills a knowledge gap and is eagerly awaited.
---

REVIEWER	Inamasu, Joji Fujita Health University Hospital, Neurosurgery
REVIEW RETURNED	05-Mar-2023

GENERAL COMMENTS	In most [spontaneous] spinal cord infarction patients, the causes are either embolic, atherosclerotic, inflammatory (vasculitis), or aortic dissection (unrelated to aortic surgery), and in rare occasions, the cause remains unknown. When we talk about stroke (cerebral infarction), we do not use the word [idiopathic] stroke; we use the word [stroke of unknown cause/source] instead. The same thing will apply to spinal cord infarction: it will better not use the word [idiopathic] (page 8), just focusing on [spontaneous] spinal cord infarction.
---

VERSION 1 – AUTHOR RESPONSE

Reviewer: 1

Prof. Johann Sellner, Paracelsus Medical University Salzburg

Comments to the Author:

The protocol for the systematic literature review on seems to consider all relevant aspects. The analysis fills a knowledge gap and is eagerly awaited.

We thank the reviewer for their appreciation of our planned work. We look forward to disseminate the final product once done.

Reviewer: 2

Dr. Joji Inamasu, Fujita Health University Hospital

Comments to the Author:

In most [spontaneous] spinal cord infarction patients, the causes are either embolic, atherosclerotic, inflammatory (vasculitis), or aortic dissection (unrelated to aortic surgery), and in rare occasions, the cause remains unknown. When we talk about stroke (cerebral infarction), we do not use the word [idiopathic] stroke; we use the word [stroke of unknown cause/source] instead. The same thing will apply to spinal cord infarction: it will better not use the word [idiopathic] (page 8), just focusing on [spontaneous] spinal cord infarction.

We thank the reviewer for this comment. We have now deleted the word idiopathic from all passages of the manuscript.